# A Model-Based Pharmacokinetic/Pharmacodynamic Analysis of the Combination of Amoxicillin and Monophosphoryl Lipid A Against *S. pneumoniae* in Mice

**DOI:** 10.3390/pharmaceutics13040469

**Published:** 2021-03-30

**Authors:** Sebastian Franck, Robin Michelet, Fiordiligie Casilag, Jean-Claude Sirard, Sebastian G. Wicha, Charlotte Kloft

**Affiliations:** 1Department of Clinical Pharmacy and Biochemistry, Institute of Pharmacy, Freie Universitaet Berlin, 12169 Berlin, Germany; sebastian.franck@fu-berlin.de (S.F.); robin.michelet@fu-berlin.de (R.M.); 2Department of Clinical Pharmacy, Institute of Pharmacy, University of Hamburg, 20146 Hamburg, Germany; sebastian.wicha@uni-hamburg.de; 3CNRS, INSERM, CHU Lille, Institut Pasteur de Lille, U1019-UMR8204-CIIL-Center of Infection and Immunity of Lille, University Lille, 59019 Lille, France; fcasilag@gmail.com (F.C.); jean-claude.sirard@inserm.fr (J.-C.S.)

**Keywords:** pharmacometric PK/PD modelling, time-to-event modelling, amoxicillin, MPLA, immunomodulation, murine model

## Abstract

Combining amoxicillin with the immunostimulatory toll-like receptor 4 agonist monophosphoryl lipid A (MPLA) represents an innovative approach for enhancing antibacterial treatment success. Exploiting pharmacokinetic and pharmacodynamic data from an infection model of *Streptococcus pneumoniae* infected mice, we aimed to evaluate the preclinical exposure-response relationship of amoxicillin with MPLA coadministration and establish a link to survival. Antibiotic serum concentrations, bacterial numbers in lung and spleen and survival data of mice being untreated or treated with amoxicillin (four dose levels), MPLA, or their combination were analyzed by nonlinear mixed-effects modelling and time-to-event analysis using NONMEM^®^ to characterize these treatment regimens. On top of a pharmacokinetic interaction, regarding the pharmacodynamic effects the combined treatment was superior to both monotherapies: The amoxicillin efficacy at highest dose was increased by a bacterial reduction of 1.74 log10 CFU/lung after 36 h and survival was increased 1.35-fold to 90.3% after 14 days both compared to amoxicillin alone. The developed pharmacometric pharmacokinetic/pharmacodynamic disease-treatment-survival models provided quantitative insights into a novel treatment option against pneumonia revealing a pharmacokinetic interaction and enhanced activity of amoxicillin and the immune system stimulator MPLA in combination. Further development of this drug combination flanked with pharmacometrics towards the clinical setting seems promising.

## 1. Introduction

The rational use of our arsenal of anti-infective drugs provides the basis of a successful treatment of bacterial infections. Unfortunately, the emergence of resistance challenges this paradigm. Treatment of lower respiratory tract infections (LRTI), currently one of the leading infectious causes of death worldwide [1] and mainly caused by *S. pneumoniae* [2], is highly affected by antibiotic resistance [1]. Due to the lack of new antibiotics, it is urgently required to investigate innovative treatment options such as stimulating the innate immune system [3].

One frequently used drug to treat LRTI is amoxicillin (AMX), a well-established beta-lactam antibiotic classified as essential medicine by the World Health Organisation [4]. To sustain the drug’s effectiveness, the immunomodulatory characteristics of monophosphoryl lipid A (MPLA) were recently studied in combination with AMX [5]. MPLA, a toll-like receptor 4 (TLR4) agonist with a favorable safety profile [6], is already licensed as adjuvant with immunomodulatory characteristics for vaccines (Fendrix^®^, hepatitis B [7]); Cervarix^®^, human papilloma virus [8]). Based on in vivo data in a murine infection model, Casilag et al. proposed that MPLA in monotherapy has the ability to stimulate the immune system, resulting in higher efficacy in terms of reduced bacterial numbers at the target sites lung and spleen as well as increased survival [5]. These effects were even more pronounced in combination with AMX [5]. However, the mechanisms of these interactions are still unknown.

By bridging knowledge from drug concentrations to bacterial numbers and ultimately survival, this study aims to integrate knowledge from multiple levels of generated data and addresses a more detailed understanding of the underlying drug interactions. To gain quantitative and mechanistic insights into the combination, it is favorable to analyze pharmacokinetic (PK) and pharmacodynamic (PD) characteristics. To accomplish this, nonlinear mixed-effects (NLME) modelling in terms of PK/PD modelling can be employed to study experimentally obtained results beyond a common descriptive analysis, allowing to amalgamate information on PK and PD levels, to integrate multiple study data and to translate results into a clinical setting within one approach. Several examples in a preclinical or clinical setting have shown the advantages of this approach [9,10,11,12,13,14,15]: PK and PD, in terms of drug concentrations and bacterial numbers or survival, are analyzed simultaneously over the entire time in a coherent framework.

The objective of this study was to evaluate the preclinical PK/PD relationship of a novel experimental antibiotic and immunostimulatory combination regimen and quantitatively define its efficacy and interactions by applying pharmacometric PK/PD modelling. Using NLME modelling not only allows to link information from drug concentrations and bacterial numbers but also enables the link to survival data. Thereby, we aimed to associate overall survival to respective model-predicted parameters by performing time-to-event (TTE) analysis.

## 2. Materials and Methods

### 2.1. Preclinical Data

The pooled dataset used for the pharmacometric analysis comprised several individually performed in vivo studies: The combination of AMX administered by oral gavage (PK study: 0.4, 14 mg/kg; PD and survival study: 0.2, 0.4, 1.2 mg/kg) and a single dose of intraperitoneally administered MPLA (2.0 mg/kg) were investigated in a murine infection model [5] of two types of mice (RjOrl:Swiss, Balb/cJRj) being infected intranasally with *S. pneumoniae* (Minimal inhibitory concentration of AMX (MIC_AMX_) = 0.016 mg/L) 12 h before treatment. Mice were either untreated or treated with AMX, MPLA or the combination as described in detail elsewhere (Appendix A, [5]). Total AMX concentrations were quantified in serum by a previously developed and validated liquid chromatography tandem mass spectrometry assay [16] including an in-study validation. Bacterial colonisation and dissemination were assessed at the target sites lung and spleen over 36 h after treatment and survival was monitored every 24 h over 14 d after infection.

### 2.2. Pharmacometric Modelling

Based on serum concentrations of AMX and log-transformed bacterial numbers at both target sites over time, NLME modelling was performed to characterize PK and PD of the untreated, mono-and combination treatments using NONMEM^®^ 7.4.1 (ICON Clinical Research LLC, Gaithersburg, MD, USA). Survival data was analyzed by TTE modelling using NONMEM^®^ as well [17].

The pharmacometric PK/PD model development process followed a sequential modelling approach (Appendix A): First, a “PK submodel” capturing AMX concentrations in serum as well as a “bacterial disease submodel” based on data of the infected but untreated group describing bacterial growth in terms of bacterial numbers in lung and spleen were developed separately. Then, a transit of bacteria from lung to spleen was allowed by implementing transit compartments. Subsequently, the PK and the bacterial disease submodels were linked with fixed population parameter estimates of the PK submodel, accounting for distribution of drugs to the bacterial target site in an “effect compartment submodel” [12,18,19]. In this way, a “disease and treatment submodel” for pneumonia in mice treated with AMX and MPLA coadministration was obtained as PK/PD model. Potential interactions of AMX and MPLA were investigated both on a PK and PD level, and potential covariates such as the type of mouse, MPLA coadministration or the administered AMX dose were studied (Appendix A). In addition, the drug effects in monotherapy as well as in combination were comprehensively studied at 36 h in lung to evaluate the effect of the combination at the end of the studies.

Finally, a “TTE model” was developed describing the survival data. A link between the pharmacometric PK/PD and TTE model was established by investigating various study characteristics (e.g., study group) as well as multiple model-predicted PK and PD parameters (e.g., the time of AMX concentrations above the MIC in serum expressed as a percentage of the dosing interval (%T_>MIC_) or bacterial numbers in spleen at 36 h after treatment) as potential covariates of the TTE model (Appendix A).

During the entire model development process, graphical (visual predictive checks (VPC), goodness-of-fit plots) and numerical (objective function value, likelihood ratio test) evaluation techniques were applied at every single step and supported by bootstrap analyses at key models to evaluate the quality of the model parameters and the predictive performance of the model as well as guide the model development process (Appendix A).

## 3. Results

Pharmacometric Modelling

The separately developed PK, bacterial disease, effect compartment and disease and treatment submodels were combined to a “pharmacometric PK/PD model” (Figure 1; Table 1; Appendix A).

The final structural “PK submodel” (Figure 1, left) for AMX comprised 106 quantified murine serum concentrations (15.1% below the lower limit of quantification (LLOQ = 0.01 µg/mL), successful in-study validation with acceptable accuracy and precision) with one to two samples per individual mouse. Within this study, total AMX concentrations were quantified and used for further analyses given the relatively low and linear protein binding (17% [20]). The submodel consisted of a two-compartment model with first-order absorption including a lag time as well as first-order linear elimination and a proportional residual unexplained variability (RUV) model and was able to describe the general trend of the AMX concentration-time profile. Given the observed shift of the maximum serum concentration (C_max_) of AMX from 0.167 h in monotherapy to 0.5 h in a combined treatment approach (Appendix A), the integration of MPLA coadministration on the AMX clearance depending on the AMX dose as covariate significantly increased the predictive performance of the model as assessed by standard model evaluation techniques. Indeed, MPLA combined with the highest AMX dose (14 mg/kg) decreased the clearance of AMX by 40.9% (73.3 mL/h) compared to monotherapy; at low dose (0.4 mg/kg), the effect was negligible (–1.17%, 123 mL/h). The mouse type (RjOrl:Swiss, Balb/cJRj) did not significantly have an impact on the model performance and, hence, was not included.

The proposed “bacterial disease submodel” (Figure 1, right), characterized by natural growth and treatment-unrelated killing and natural death kinetics, described bacterial growth of intranasally administered *S. pneumoniae* in lung with bacterial growth after a slight short-term reduction in bacterial numbers adequately (Figure 2A, “AMX-/MPLA-“ study group). By use of a transit compartment model, of which the number of compartments was optimized to be 23, the transit of bacteria from lung to spleen was described. In the spleen, the bacterial input was modelled as a transit of bacteria from the lung without any outflow leading to accumulation of bacteria after more than 12 h after infection as seen in the original study data. The PK and the bacterial disease model were linked via two separate effect compartments in lung and spleen assuming a transfer of AMX to the respective organs (Figure 1, middle).

According to the reported results of the murine infection model [5], AMX monotherapy decreased bacterial numbers in the lung up to 24 h and a dose-dependent regrowth was observed afterwards, whereas MPLA-treated animals displayed more or less stable bacterial numbers after initial bacterial killing and the combination of both drugs led to the highest reduction in bacterial numbers (Symbols in Figure 2A, “AMX+/MPLA-“, “AMX-/MPLA+“ and “AMX+/MPLA+“ study groups). In the spleen, bacterial numbers increased after more than 2 h showing comparable patterns regarding the efficacy of mono- versus combination therapies as in the lung for all treatment groups (Symbols in Figure 2B). These trends were captured in the “disease and treatment submodel”, where the effect of AMX was best characterized in lung by a sigmoidal maximum effect (E_max_) model as drug-dependent bacterial killing process. The concentration-effect relationship was steep (Hill factor = 20) indicating that AMX was only efficacious as long as AMX concentrations in lung were above the AMX concentration to achieve half maximum effect (EC_50_ = 0.0109 µg/mL). In the spleen, the AMX killing effect was described by a power model. The effect of MPLA on bacteria in the lung was implemented as its ability to enhance the efficacy of the treatment-unrelated killing and natural death effects on bacteria. Contrarily, the effect of MPLA in spleen was included as separate killing process. A potential PD interaction of AMX and MPLA was further analyzed to define potential synergy or antagonism. However, the results clearly indicated an additive effect of AMX and MPLA and did not hint at any synergism or antagonism. Final model parameter estimates described the underlying PD processes reliably with acceptable relative standard errors (RSE<46.2%) and based on performed bootstrap analyses (Table 1) and VPC adequately captured measured AMX serum concentrations (Appendix A) and bacterial numbers in lung and spleen (Figure 2) above and below the LLOQ (general trend, variability, and fraction, respectively). Here, PK parameters were fixed to the final parameters of the PK submodel, although a simultaneous modelling approach of PK and PD was aimed for but excluded due to (i) plausibility (PK and PD were not determined within one animal) and (ii) to improve model stability. Further results of all model specifications are given in Appendix A.

AMX as monotherapy at the highest investigated dose (1.2 mg/kg, PD studies) and MPLA reduced model-predicted bacterial numbers in the lung by 3.03 and 1.71 log10 colony forming units (CFU)/lung, respectively, compared to natural growth after 36 h (Appendix A, left). The combination also showed a bactericidal effect with a reduction of 4.77 log10 CFU/lung. In the spleen, comparable characteristics were observed: Here, the maximum effect of total bacterial elimination was determined for doses of 0.4 and 1.2 mg/kg AMX with MPLA coadministration (Appendix A, right).

Survival studies of infected mice revealed the highest mortality for untreated mice, whereas combined treated mice had highest survival rates [5]. The “TTE model” revealed the best predictive performance of the survival data by using a surge function to describe the hazard being mostly present between day 3 and 6. Combining %T_>MIC_ in serum and coadministration with MPLA displayed the best relation and plausibility between PK, PD, and survival, when included exponentially as covariates on the overall hazard: In all study groups, survival was predicted reliably except for mice treated with 0.40 mg/kg AMX and MPLA. Here, survival was slightly overestimated, but as median observed survival was still within the 90% CI of the median simulated survival, the predictions were thus considered appropriate. Other investigated covariates, e.g., the model-predicted bacterial number in spleen after 36 h after treatment, were excluded due to the simplicity of %T_>MIC_, although a comparably good predictability was observed. VPC (Figure 3) as well as a bootstrap analysis (Table 2) supported this structure with stable parameter estimates.

A high hazard was primarily present between day 1 and 6 with the highest maximum hazard between day 3 and 5 justifying a surge function (Appendix A). A larger %T>MIC of AMX as well as presence of MPLA decreased the overall hazard and thereby increased survival: The overall hazard of untreated mice was 3.71-fold higher than for MPLA-treated mice. Most important, cotreatment with MPLA was able to reduce the hazard of AMX-treated mice at maximum dose by a factor of 4.00. After 14 d, survival was increased 1.35-fold from 66.7% for AMX monotherapy at highest dose to 90.3% in a combined treatment approach due to the immune system stimulation effect of MPLA: Coadministration of MPLA (T_>MIC_ > 3.25 h) reduced the required T_>MIC_ to reach survival >95% by 1.23 h compared to AMX monotherapy at highest dose (T_>MIC_ > 4.48 h).

## 4. Discussion

Aiming to develop novel therapeutic approaches to overcome the emergence of bacterial resistance, the PK/PD relationship of the innovative treatment option of the antibiotic AMX and the TLR4 agonist MPLA was exploited. Here, pharmacometric approaches were comprehensively investigated as useful tools to further quantitatively interpret in vivo defined outcomes.

Experimental data indicated beneficial effects of AMX, when coadministered by MPLA to stimulate the immune system, but lacked a more pronounced mechanistic and quantitative understanding of involved processes and interactions [5]. Our pooled pharmacometric analysis of various studies integrated several information starting from PK in terms of drug concentrations to PD in terms of bacterial numbers and survival to provide quantitative and mechanistic insights into this treatment option against pneumonia. On basis of the comprehensive analysis of the immune system stimulating effects of MPLA and the antibiotic effects of AMX by Casilag et al. [5], we demonstrated that MPLA has the ability to stimulate the immune system and additively enhanced the AMX activity on top of a PK interaction. We stepwise integrated more data allowing reliable estimation of PK/PD relationships that were successfully linked to overall survival employing easily accessible clinical parameters (%T_>MIC_). This analysis highlights the necessity to amalgamate multiple levels of generated data to gain detailed information of underlying interactions and physiological processes.

In our “PK submodel”, we found that MPLA substantially reduced the AMX clearance depending on the AMX dose. Although no pharmacokinetic studies analyzing the combination of AMX and MPLA have been published, an analogous pharmacometric model has been reported by Moine et al. in mice being treated with AMX subcutaneously [21] supporting plausibility of the “PK submodel”. The elimination of AMX monotherapy with a clearance of 124 mL/h was higher than reported creatinine clearance values in mice [22,23] indicating tubular secretion of AMX, most probably by organic anion transporters [24]. This is also in agreement with described elimination processes of AMX being rarely non-renal [20]. MPLA is also partly excreted renally [6] and one can hypothesize that MPLA reduces tubular secretion of AMX (PK interaction) competitively at high AMX concentrations. Such a mechanism may also result in a potential PK interaction in humans. Therefore, a clinical dose finding study and drug–drug interaction study would need to be conducted to evaluate the relevance in humans. The identified PK interaction is limited given the fact that only a standard dose of MPLA, that was chosen based on prior studies [5], and two different doses of AMX were studied, and hence the PK interaction was simplified by a linear relationship, which, e.g., could also have been exponential or sigmoidal. Still, the PK interaction did not quantitatively explain the PD results after combination therapy, especially at the investigated relatively low doses of the PD study (0.2–1.2 mg/kg) that only displayed negligible PK interaction compared to the highest dose of the PK study (14 mg/kg). Consequently, a PD interaction was investigated by analyzing bacterial numbers on top of the PK interaction.

The “bacterial disease submodel” in the lung consisted of a first-order growth process being influenced by a delay in onset of bacterial growth probably because bacteria needed to adjust to the new environment. The high model-predicted number of transit compartments characterized the manifold involved processes needed for the transit of bacteria from lung to spleen and displayed the physiological relevance: Several membranes have to be passed by bacteria to enter the blood stream and ultimately reach the spleen. The implemented bacterial elimination contained not only natural death of bacteria, but also killing effects attributable to the immune system that were predicted to be present consistently over time after initial stimulation by the bacteria. One limitation of our analysis is that we were not able to distinguish these processes within the model, which could, e.g., be investigated in immunodeficient mice in a next step. Nevertheless, bacterial growth outperformed the killing processes over time leading to a higher bacterial burden and, subsequently, reduced survival rates.

Analyzing the “disease and treatment submodel”, the killing effects of AMX and MPLA were successfully characterized. Separate first-order rate constants for the effect delay of AMX in lung and spleen were determined. These empirically derived differences seem to be plausible given the physiological differences in the AMX transfer from serum to lung and spleen as a manifold of processes are involved, e.g., different membranes, transporters, or blood flow. AMX bacterial killing in the lung was best described by on–off kinetics with the EC_50_ as threshold being related to effect compartment concentrations, whereas MPLA increased the immune system attributable killing effects constantly over time, showing highest killing in combination with AMX. Although the AMX effect was rather expected to be drug than organ specific, the effects of AMX needed to be modelled differently between lung and spleen. The different situation in the lung compared to the spleen, where a slower equilibrium formation with serum (Appendix A) led to longer lasting effects and missing observations attributable to killing effects of the immune system, required a more simplified implementation in the spleen and led to a missing bacterial outflow from the spleen compartment. Here, future studies would benefit from quantifying drug concentrations at the target sites to describe these processes more physiologically. The model-predicted efficacy of AMX being associated with a high time of antibiotic concentrations above the EC_50_ (T_>EC50_) correlated well with the %T_>MIC_, the main PK/PD parameter used for the usually time-dependent beta-lactam antibiotics. %T_>MIC_ as an easily accessible clinical parameter in addition to simple binary MPLA implementation was able to capture murine survival after infection.

In this preclinical setting, mice received only a single dose of AMX. Our results suggest that AMX affects bacteria within the first phase of infection, mainly visible only for a limited period, whereas MPLA at the same time stimulates the immune system in a sustained manner. Unfortunately, the PK of MPLA was not monitored due to missing sensitivity of investigated MPLA assays, and, hence, no response surface analysis [10,25] was feasible to comprehensively investigate possible further PD interactions over time. However, the immune system stimulating characteristics of MPLA may be advantageous in a clinical framework.

This work adds weight to the use of pharmacometric PK/PD modelling and subsequent survival analysis being useful tools to support and further quantitatively interpret experimentally defined outcomes of in vivo studies and to evaluate the effectiveness of the proposed combination comprehensively. Despite certain limitations such as the variability in the experimental data or the ethical limitation of only one sample per individual mouse in terms of limited blood volume (PK) or physiological limitations (PD) that only allowed to analyze the in vivo data with caution due to missing intraindividual variability especially for PD observations, valuable quantitative insights into the combination were achieved by the use of NLME modelling taking all these different individual observations into account.

The herein outlined approach contributes to reducing the use of animals in preclinical drug development studies, since various scenarios, e.g., investigating different dosing regimens, can be simulated (a priori). In a next step, the PK/PD-TTE model is apt to be translated into a clinical setting for the clinical development program bringing this promising combination therapy closer to patients as translation to clinics of comparable approaches based on immune stimulation has already been proposed [3]. However, due to the use of separate mice for PK, PD and immunostimulation experiments, translating the current results to humans is challenging. To accomplish this step, additional concentration-effect data, or dose-response data for MPLA would be needed. Therefore, further studies investigating adjunct use of immunostimulatory compounds should use i) different doses of the immunomodulator, ii) measure PK and PD simultaneously at least in some experiments, and iii) use multiple experimental systems such as different animal species, immunocompetent and -deficient animals or in vitro set-ups tailored to specific endpoints of interest. Still, the safe use of MPLA in humans and the augmented innate antimicrobial immunity might establish a basis for a successful translation as adjunct therapy to conventional antibiotic treatments.

In conclusion, we demonstrated that combining antibiotics with immunostimulatory drugs may serve as a promising approach to augment the effect of antibiotics. Leveraging a pharmacometric PK/PD analysis approach, the performed analysis showed the ability of MPLA to stimulate the immune system and additively enhance the efficacy of the antibiotic AMX in terms of bacterial burden and survival. Further studies with other antibiotics and—ultimately—evaluation of such approaches in the clinical setting are warranted.

## Figures and Tables

**Figure 1 pharmaceutics-13-00469-f001:**
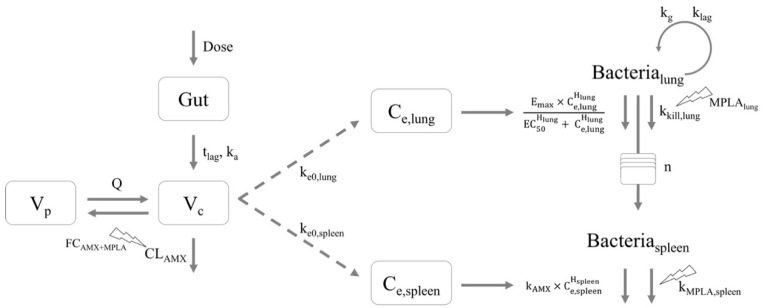
Schematic illustration of the pharmacometric nonlinear mixed-effects pharmacokinetic/pharmacodynamic model for amoxicillin (AMX) with coadministration of monophosphoryl lipid A (MPLA) of mice infected with *Streptococcus pneumoniae* serotype 1, comprising a two-compartments pharmacokinetic (PK) model (**left**), an effect compartment as PK/pharmacodynamic (PD) link model (**middle**) and a disease and treatment model including unrelated killing and natural death effects including killing effects of the immune system, and killing effects of AMX and MPLA (**right**). Abbreviations: Bacteria_lung_: Number of bacteria in lung; Bacteria_spleen_: Number of bacteria in spleen; C_e,lung_: AMX concentration in lung effect compartment; C_e,spleen_: AMX concentration in spleen effect compartment; CL_AMX_: Clearance of AMX; Dose: Administered dose of AMX by oral gavage; EC_50_: Concentration of AMX to achieve half maximum killing effect; E_max_: Maximum killing effect of AMX; FC_AMX + MPLA_: Fractional change of CL_AMX_ in presence of MPLA depending on the AMX dose implemented as covariate; Gut: Organ of AMX administration; H_lung_: Hill factor for lung; H_spleen_: Hill factor for spleen; k_a_: First-order absorption rate constant; k_AMX_: First-order killing rate constant of AMX in spleen representing the slope of the effect compartment concentration and effect relationship; k_e0,lung_: First-order rate constant for effect delay in lung; k_e0,spleen_: First-order rate constant for effect delay in spleen; k_g_: First-order growth rate constant in lung; k_kill,lung_: First-order rate constant for treatment-unrelated killing and natural death effects in lung; k_lag_: First-order rate constant for delay in onset of bacterial growth in lung; k_MPLA,spleen_: First-order killing rate constant for killing effect in spleen only in presence of MPLA; MPLA_lung_: Fractional change of k_kill,lung_ in presence of MPLA; n: Number of transit compartments; Q: Intercompartmental clearance; t_lag_: Lag time; V_c_: Central volume of distribution; V_p_: Peripheral volume of distribution.

**Figure 2 pharmaceutics-13-00469-f002:**
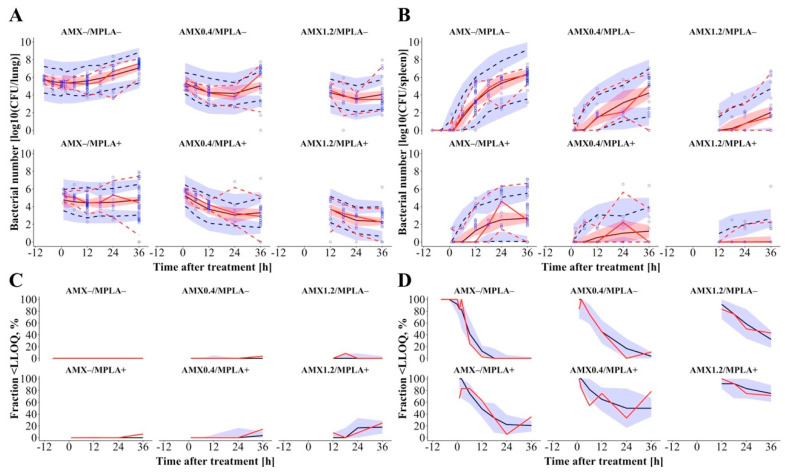
Visual predictive check (n = 1000 simulations including unexplained variability) of the pharmacometric pharmacokinetic/pharmacodynamic model for bacterial numbers in lung (**A**) and spleen (**B**) stratified into study groups including the fractions of samples being below the LLOQ for lung (**C**) and spleen (**D**). Circles: Observations; Lines: 50th percentile (solid), 5th and 95th percentile (dashed) of observed (red) and simulated (black) bacterial numbers. 90% confidence interval around simulated percentiles as shaded area. Abbreviations: AMX: Amoxicillin (0.40 mg/kg or 1.20 mg/kg); LLOQ: Lower limit of quantification; MPLA: Monophosphoryl lipid A (2.00 mg/kg); +: Treatment with respective drug; –: No treatment with respective drug.

**Figure 3 pharmaceutics-13-00469-f003:**
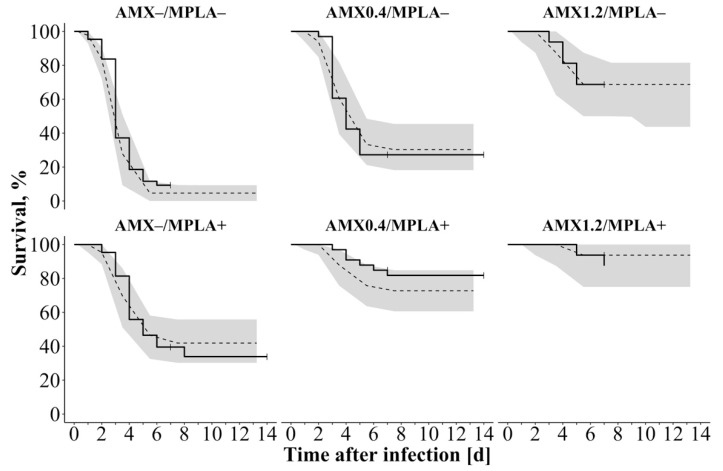
Visual predictive check of the overall survival model (n = 1000 simulations including unexplained variability) for different study groups of mice infected with *Streptococcus pneumoniae* serotype 1 and untreated or treated with AMX with or without coadministration of MPLA: Solid lines: Observed survival; Dashed lines: Simulated survival; 90% confidence interval around simulated survival as shaded area. Abbreviations: AMX: Amoxicillin (0.40 mg/kg or 1.20 mg/kg); MPLA: Monophosphoryl lipid A (2.00 mg/kg); +: Treatment with respective drug; –: No treatment with respective drug.

**Table 1 pharmaceutics-13-00469-t001:** Model-predicted parameter estimates including bootstrap results (convergence rate of 99.7%) of a sequential analysis of the pharmacometric pharmacokinetic/pharmacodynamic model of amoxicillin and monophosphoryl lipid A.

Parameter	Parameter Estimate	Bootstrap	
**[unit]**	Estimate (%RSE)	Median	95% CI
**Pharmacokinetic Model**			
k_a_ [h^−1^]	5.04 *	5.04 *	-
t_lag_ [h]	0.125 *	0.125 *	-
V_c_/F [mL]	15.4 *	15.4 *	-
V_p_**/**F [mL]	50.7 *	50.7 *	-
Q/F [mL/h]	71.9 *	71.9 *	-
CL_AMX_/F [mL/h]	124 *	124 *	-
FC_AMX+MPLA_ [mL/h/µg]	−0.145 *	−0.145 *	-
**Effect Compartment Model**			
k_e0,lung_ [h^−1^]	0.125 (19.7)	0.125	0.0830–0.247
k_e0,spleen_ [h^−1^]	0.0435 (17.7)	0.0435	0.0287–0.0635
**Bacterial Disease Model**			
N_bacteria, t=0_[log10(CFU/lung)]	6.12 **	6.12 **	-
k_g_ [h^−1^]	0.477 (7.00)	0.504	0.439–1.12
k_lag_ [h^−1^]	0.0595 (46.2)	0.0555	0.0111–0.108
k_kill,lung_ [h^−1^]	0.274 (20.3)	0.270	0.190–0.382
n	23.0 (13.1)	23.3	18.0–31.4
MTT [h]	40.8 (4.30)	40.5	37.2–44.5
**Disease and Treatment Model**			
MPLA_lung_	1.40 (6.10)	1.41	1.29–1.61
E_max_ [h^−1^]	0.255 (6.00)	0.253	0.220–0.283
EC_50_ [µg/mL]	0.00109 (29.4)	0.00109	0.000134–0.00146
H_lung_	20 **	20 **	-
k_MPLA,spleen_ [h^−1^]	3.71 (27.5)	3.50	2.05–6.12
k_AMX_ [log10(h^−1^)]	13.7 (22.1)	14.0	9.05–24.2
H_spleen_	5.06 (23.9)	5.14	3.22–9.26
**Residual Unexplained Variability *****			
Lung [log10(CFU/lung)]	1.12 (3.90)	1.12	1.03–1.21
Spleen [log10(CFU/spleen)]	1.81 (4.40)	1.79	1.65–1.95

Abbreviations: AMX: Amoxicillin; CI: Confidence interval; CL_AMX_: Clearance of AMX; EC_50_: Concentration of AMX to achieve half maximum effect; E_max_: Maximum effect of AMX; F: Bioavailability of AMX fixed to 1; FC_AMX+MPLA_: Fractional change of CL_AMX_ in presence of MPLA depending on the AMX dose implemented as covariate; H_lung_: Hill factor in lung; H_spleen_: Hill factor in spleen; k_a_: First-order absorption rate constant; k_AMX_: First-order killing rate constant of AMX in spleen; k_e0,lung_: First-order rate constant for effect delay in lung; k_e0,spleen_: First-order rate constant for effect delay in spleen; k_g_: First-order growth rate constant of bacteria in lung; k_kill,lung_: First-order rate constant for treatment-unrelated killing and natural death effects in lung; k_lag_: First-order rate constant for delay in onset of bacterial growth in lung; k_MPLA,spleen_: First-order killing rate constant for killing effect in spleen in presence of MPLA; MPLA: Monophosphoryl lipid A; MPLA_lung_: Fractional change of k_kill,lung_ in presence of MPLA; MTT: Mean transit time; n: Number of transit compartments; N_bacteria, t=0_: Initial number of bacteria in lung at −12 h; Q: Intercompartmental clearance; RSE: Relative standard error; t_lag_: Lag time; V_c_: Central volume of distribution; V_p_: Peripheral volume of distribution; * Fixed parameter estimates of developed PK submodel (Appendix A); ** Fixed parameter estimate based on model development process (sensitivity analysis, log-likelihood profiling and bootstrap results) due to stability and plausibility (Appendix A); *** Residual unexplained variability estimated on standard deviation scale.

**Table 2 pharmaceutics-13-00469-t002:** Model parameter estimates including bootstrap results (convergence rate of 100%) of the time-to-event analysis of survival data of mice untreated or treated with amoxicillin and/or monophosphoryl lipid A.

Parameter	Parameter	Bootstrap	
**(unit)**	(%RSE)	Median	95% CI
**Structural base model**			
SA (h^−1^)	0.0404 (20.9)	0.0413	0.0263–0.0676
SW (h)	35.7 (17.3)	36.0	23.5–67.1
γ	2.24 (26.0)	2.36	1.38–7.39
PT (h)	89.2 (4.80)	89.4	81.6–110
**Covariate model**			
T_>MIC_	−0.926 (16.7)	−0.940	−1.28 to (–0.654)
MPLA_TTE_	−1.32 (16.6)	−1.33	−1.82 to (–0.878)

Abbreviations: γ: Shape parameter; CI: Confidence interval; MPLA_TTE_: Covariate effect of coadministration of monophosphoryl lipid A on overall survival; PT: Peak time; SA: Surge amplitude; SW: Surge width at half maximum intensity; T_>MIC_: Time of amoxicillin concentrations in serum above the minimal inhibitory concentration.

## Data Availability

Data can be received from the authors upon reasonable request.

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
