# Peer review of "A Model-Based Pharmacokinetic/Pharmacodynamic Analysis of the Combination of Amoxicillin and Monophosphoryl Lipid A Against S. pneumoniae in Mice"

_pharmaceutics, 2021, doi:10.3390/pharmaceutics13040469_

Round 1

Reviewer 1 Report

Overall, this study provided a comprehensive description and clear result to illustrate the combination treatment of amoxicillin and monophosphoryl lipid A 3 against S. pneumoniae in mice by PK/PD modeling approach. I appreciate the effort of the authors and would like to provide some comments to improve this study. The main issue is modeling. Even though the process is clearly describing in the article, I'd like to ask the author to provide their source code in supplementary. Is it possible that the author can provide the piece of code to demonstrate how they conducted the modeling and analysis process? It would be helpful to the reader to have a clear and transparent M&S process in this study.   Some minor question/comments are list below,   Table 1. The way to represent the estimation of parameters is easy to confuse. The median should place before %RSE. Table 1. Why there is no 95%CI for PK model parameters? Table 1. Why the unit of Emax in the treatment model is h^-1. I think Emax is not a rate constant in your model. Table 1. What is the residual unexplained variability? How did the author obtain these values in this study? Figure 2. It doesn't make sense why the author simulated the 90% CI around the percentile. What is the purpose of these shade areas? I'd like the figure is easy to read and compare the observation and prediction (e.g., using the points as observation and line with shade as prediction). Figure 3. It seems the survival rate is overestimated in AMX0.4/MPLA+. Should add some discussion about this issue.

Reviewer 2 Report

Sebastian Franck reported a model-based PK/PD analysis of the combination of amoxicillin and monophosphoryl lipid A against S. pneumoniae in mice. The experiments are well designed, and the data are clearly interpreted and presented. I only have a few comments. The authors are suggested to discuss the rationale of using nonlinear mixed-effects (NLME) and time-to-events (TTE) modeling to analyze the PK/PD data. The authors are also suggested to discuss the rationale of dosing selection in the PK and PD studies.

Reviewer 3 Report

The author’s have submitted a manuscript describing a PK-PD analysis of amoxicillin + MPLA combination administered to mice inoculated with S. pneumoniae.  The insurgence of antibacterial resistance is a significant health burden, as the authors point out. One option to surmount this problem is to add another agent to an already established antibiotic, a regimen considered in the present paper. Overall, I found the article well and thoroughly written. A drawback to the paper is the authors lack of providing sufficient details on the PK interaction – this is certainly an interesting finding and how this may affect clinical care is not discussed. Also, the methods section in the article file is too short, so more details are necessary, rather than just referring to the supplemental file as is done in the current version (from a reader’s perspective this is frustrating when there is lack of information and another file has to be downloaded, read, etc.). I have several other comments:

  1. A brief description of why the animal species were chosen is needed.
  2. Why were the doses for PK and PD studies different except one dose (0.4 mg/kg)?
  3. I was surprised to read that the LLOQ of the LC-MS assay was only 10 ng/mL. With sophisticated equipment such as a mass spectrometer it is routine to achieve LLOQ less than 1 ng/mL. Also, this LLOQ is close to the MIC of amoxicillin (16 ng/ml as stated in the article). It would be helpful for the authors to discuss this in lieu of the present conclusions.
  4. A reference and more details are required to describe the assay that determined CFU.
  5. For studies employing destructive sampling, a large number of animals (e.g. >10 animals per time point) are needed to provide significant results. The authors should consider this when interpreting the results.
  6. To increase the clinical relevance of the findings, that authors might consider using the terms “bactericidal” and “time-dependent antibiotic” at various places in the manuscript.
  7. PK Interaction – can the authors speculate, based on the findings, about the mechanism by which MPLA prolongs time to Cmax of amoxicillin? MPLA was administered by i.p. injection and amoxicillin was administered by oral gavage.  Would you anticipate a gut interaction?
  8. PK Interaction – It is surprising that the authors do not provide a table that displays the PK results across treatment groups. A reduction in amoxicillin clearance by MPLA may be clinically relevant as build up of amoxicillin would be a concern in patients with compromised renal function.
  9. The first paragraph in the Discussion section (pg. 8) should be re-phrased or separated into multiple sentences.
  10. 8, line 288-290. The authors incorrectly state that their findings show that MPLA induces immune function – which immunological parameters are you comparing to support this claim?  It is true that the combination appears to have greater kill but this may be due to the drug interactions irrespective of the immune system.  I have the same issue with the statement on pg. 9, lines 346-347.
  11. The authors should compare the PK results in the present paper to those published earlier. Are there any similarities or discrepancies?
  12. The article lacks a Conclusion section and thus should be added.
